# Water as a Link between Membrane and Colloidal Theories for Cells

**DOI:** 10.3390/molecules27154994

**Published:** 2022-08-05

**Authors:** E. Anibal Disalvo, A. Sebastian Rosa, Jimena P. Cejas, María de los A. Frias

**Affiliations:** Applied Biophysics and Food Research Center (Centro de Investigaciones en Biofisica Aplicada y Alimentos, CIBAAL, Laboratory of Biointerphases and Biomimetic Systems, National University of Santiago del Estero and CONICET), RN 9-Km 1125, Santiago del Estero 4206, Argentina

**Keywords:** lipid hydration, water interphases, crowded systems, restricted environments, H bonding propagation

## Abstract

This review is an attempt to incorporate water as a structural and thermodynamic component of biomembranes. With this purpose, the consideration of the membrane interphase as a bidimensional hydrated polar head group solution, coupled to the hydrocarbon region allows for the reconciliation of two theories on cells in dispute today: one considering the membrane as an essential part in terms of compartmentalization, and another in which lipid membranes are not necessary and cells can be treated as a colloidal system. The criterium followed is to describe the membrane state as an open, non-autonomous and responsive system using the approach of Thermodynamic of Irreversible Processes. The concept of an open/non-autonomous membrane system allows for the visualization of the interrelationship between metabolic events and membrane polymorphic changes. Therefore, the Association Induction Hypothesis (AIH) and lipid properties interplay should consider hydration in terms of free energy modulated by water activity and surface (lateral) pressure. Water in restricted regions at the lipid interphase has thermodynamic properties that explain the role of H-bonding networks in the propagation of events between membrane and cytoplasm that appears to be relevant in the context of crowded systems.

## 1. Introduction


*The air-water interface of a drop of water has its specific heat and hence its entropy potential, i.e., an entropy potential independent of the bulk. A perturbation of such an interface is conserved. This inevitably leads—due to the conservation of momentum and the laws of thermodynamics—to propagation phenomena.*

*(Einstein, 1901)*


According to cell membrane theory, membranes form the boundary of living cells and regulate transport in and out of the cell by providing a dynamic barrier between the cellular constituents and the extracellular environment. Within this paradigm, the basic structure of the biological membrane is the lipid bilayer. The interactions between the constituent lipid molecules are at expense of water which plays a major role in the membrane properties such as stability, permeation and related functions [1].

The membrane theory has received most of the attention from physiologists and biophysicists due to the acceptance of the Fluid Mosaic Membrane (FMM) model formulated originally by Singer and Nicholson [2] and modified by others [3,4].

All of them take the lipids as the responsible for forming a semipermeable membrane encapsulating the cytoplasm which is justified from an evolution point of view with the argument that one of the first steps in life is the formation of closed supramolecular structures confining a reactive electrolytic media [5,6].

Opposed to this view, the Association Induction Hypothesis (AIH) describes a cell as a colloidal coacervate of gel proteins in water in which the protoplasm responds to metabolic events through changes in protein conformation [7,8].

While the cell membrane theories mostly ignore water as part of the structure and the thermodynamic (functional) properties derived from it, the AIH put emphasis on water ignoring membranes and lipids. However, both theories contain relevant concepts to understand cell physiology, which are in certain aspects, complementary to a unified cell theory considering its functionality as the final goal.

For this purpose, the unique properties of water linked to membranes appear to be a way by which the two proposals may converge. This requires a deeper knowledge of water structure and hydration in biological membranes. Specifically, its role in surface phenomena in different lipid assemblies and in contact with crowded systems. For this purpose, a new view of the membrane in which the interphase region is taken as an open bidimensional solution of lipid head groups is discussed.

With this aim, this article deals with the following concepts:-A brief review of the membrane and coacervate theories

*The membrane theory*.


*The coacervate theory*


-The membrane interphase model.-The membrane as an open system. Thermodynamic consequences.-Membrane hydration and membrane state.-The surface domains. Excluded volume concept.-Critical water activity and the cut-off surface pressure

*Critical states: packing and cut off pressure*.


*Cut off and critical water activity.*


-Hydration and phase transition.-The limits of the membrane in crowded systems and signal propagation.-Unifying membrane approach with water colloid systems.-The thermodynamic response: membrane state/hydration state/state- function relationship.


*Responsive structures and H bond networks.*


-Concluding remarks: Membrane hydration and membrane state. An alternative way to membrane response.

## 2. A Brief Review of the Membrane and Coacervate Theories

### 2.1. The Membrane Theory

The classical paradigms supporting the actual membrane models are resumed below.

The lipid bilayer is the backbone of the membrane in which proteins can be inserted. In some conditions, non-lamellar aggregates can be formed as transient structures depending on the lipid composition [1].

The lipid bilayer conformation is a selective permeability barrier in which water and non-polar solutes can cross driven by a concentration gradient. Permeability is evaluated as a partition-diffusion process in which the lamellar conformation is not altered. Under this paradigm, solutes dissolve in the membrane and water copermeates. On the other hand, ions and most polar solutes cannot permeate the lipid membrane, and therefore, it occurs due to the presence of specialized proteinaceous carriers or channels coupled to metabolism (active transport). The cell is filled with ordinary water with small solutes including K^+^ in solution. To compensate for the passive leaks, ion pumps located at the membrane continuously operate, a process that is considered energetically impossible [8].

Under this view, ions and other biocompounds such as aminoacids are excluded from membrane bulk due to their low solubility in the hydrocarbon region [9]. However, experimental evidences have shown that water can be found as pockets in the membrane structure favoring the permeation of some polar aminoacids [10,11]. Thus, the classical concept of permeability in which the bilayer is considered as a hydrocarbon slab where the dielectric constant is 2 has strong limitations.

Based on the concept of the lipid assemble as a phase, the bilayer suffers structural changes induced by temperature and water content by which permeability, area per lipid, and thickness are drastically modified. The phase transition is mainly ascribed to the fusion process of the hydrocarbon region in fully hydrated membranes.

The current models focus on the presence of different lipid lateral arrangements in the membrane plane (domains) due to the heterogeneity in composition. Some lipids in their pure form can stabilize in water forming non-lamellar structures such as phosphatidylethanolamines (PE) and glyceryl monooleate (GMO) [12,13]. Therefore, it has been speculated that when those lipids are present in the membrane, non-bilayer structures may be formed and therefore explain changes in permeability to ions and polar solutes. In these conditions, some lipids may act as ionophores [14]. However, the phenomena seem to be an all or none process, in which selectivity, specificity for ions, and gradual modulation are not described. In all cases, little or no role of the water surface state of the membrane has been considered.

The proposal of membrane theory relies on rules mainly deduced for bulk phases large enough to neglect interfacial phenomena. Under this view, permeation occurs driven by a chemical potential difference of the permeant between the bulk phases on both sides of the membrane. Thus, permeability is interpreted in terms of the Henry law (partition of a single solute between bulk water and the hydrocarbon region without water). In addition, diffusion is considered to be governed by Fick’s law, meaning that the diffusion coefficient is constant during the process.

The solubility-diffusion theory was questioned introducing the role of the polar head group arrangements in a three-layer theory that incorporates the area dependence as a primary modulating parameter [15]. This theory implicitly considers the presence of water in the bilayer structure.

For convenience, bilayers (liposomes, vesicles) and monolayers (extended on the air-water surface) have been extensively used as experimental model systems in a nearly independent way. Monolayers’ behavior was mostly analyzed considering that lipids behave as a van der Waals gas spread on the water surface [16,17]. The water/lipid interaction is not taken into account and, consequently, the membrane is described as a closed system in thermodynamic equilibrium with the adjacencies. On the other hand, bilayers are modeled as fully hydrated lipids in which lateral pressure cannot be controlled [17,18].

From the phenomenological point of view, an extensive discussion has led to some consensus on the conditions in which monolayers may be considered equivalent to a bilayer of the same lipid species. The bilayer equivalent pressure is accepted to be at around Π = 30 mN/m at which the phospholipase A_2_ activity is similar in both systems [19,20,21]. Estimation of the bilayer equivalent pressure is purely theoretical because bilayer lateral pressure is not experimentally measurable. Another frequently cited point is the bilayer equivalent molecular area, which for DPPC bilayers is ~64 Å^2^, which is coincident with that determined in monolayers at a saturation point [22,23].

### 2.2. The Coacervate Theory

The coacervate theory was first proposed by the Russian biochemist A.I. Oparin in 1936 [24]. According to it, the origin of life was preceded by the formation of mixed colloidal units called ‘coacervates’. These are particles composed of two or more colloids which might be proteins, lipids, or nucleic acids.

Under the view of this theory, the cell is described as a colloid with distribution coefficients and adsorption coefficients as prime physical-chemical parameters allowing a negative-entropy driven bioenergetics based on coherence [25].

Ling developed a complete colloid model for the living cell, the so-called ‘association-induction-hypothesis’ (AIH), which is claimed to be able to explain the coherent behavior of cells without the need to invoke the presence of the membrane [26].

In short, the membrane theory favors the idea that a cell is a solution of proteins while the coacervate view considers the cell as proteins dissolving water.

In this review, the consideration of water in the membrane extending the interphase to the cell interior gives the possibility to reconcile both views in terms of considering the cell as a crowded system [27].

## 3. The Membrane Interphase Model

The point that the bilayer/monolayer system cannot be appropriately treated with laws defined for bulk macroscopic systems derives from the need to consider new paradigms introducing concepts of surface physical chemistry [28]. This new view is based on Einstein’s words in the heading of this article. Moreover, other physical chemists, namely Guggenheim, Defay and Prigogine among others have called attention to the particular phenomena of interphase, but they have not been explicitly incorporated into the classical membrane literature.

The common factor behind the redefinitions of membrane properties within the new paradigm of physical chemistry of surfaces is that water is not considered as part of the membrane structure. Therefore, to include it, it is necessary to examine experimental evidence concerning water in membranes and therefore to choose the right thermodynamic approach in relation to surfaces.

There are, at least, three ways to describe the surface phenomena (Figure 1). The first, defined by Gibbs, considers the interface as an ideal plane separating two milieus of different physical-chemical properties (Figure 1A). This definition, when applied to the membrane, takes into account the hydrocarbon core framed by the polar regions being of similar properties than bulk water. In lipid membranes, the interface corresponds to the plane running along the glycerol backbone in which carbonyl groups can be oriented either towards the membrane or to the aqueous phase. This definition gives importance to the separation of the non-polar region from the polar one, without giving relevance to the physical-chemical properties of the head polar group region. Within this approach, the capacitance and the thickness of the bilayer have been calculated. Although the thickness of the bilayer does not fit a pure hydrocarbon slab, which suggests a more complex structure in terms of dielectrics; most biophysical studies use this interface idea [29].

In another definition, provided by Guggenheim, two phases of quite different polarities are separated not by a plane but by a finite region (called the interphase) whose properties differ from the bulk phase which is in contact (Figure 1B). This region has quite different properties in comparison with the two pure bulk phases in contact. It is, to some extent, considered as a mixture of the two components although the exact ratios are not easy to assess.

The Guggenheim model is particularly appropriate to adapt to a lipid membrane if the contribution of the head group region is taken into account (Figure 1C). In connection with this definition, the thermodynamics of a monolayer was described by Defay-Prigogine (and extended by Damoradam and Disalvo) as a bidimensional solution of polar head groups embedded in water [30,31,32]. This visualization of the interphase makes that monolayers’ and bilayers’ behavior can be compared considering the physical chemistry of this region as an aqueous polar solutions, something that in the current biological literature is absent [33].

The principal feature of the interphase is that it is conformed by an aqueous solution of hydrated head groups that can be treated as ions in the solution. Thus, the physical chemistry of that region can be reasonably evaluated in terms of surface solution components (head groups, ions, and water) in a bidimensional arrangement.

A direct consequence of this lipid membrane model is that water becomes a crucial component of the membrane structure that defines its thermodynamic state.

In terms of structure, it gives a more realistic picture since the microscopic water organization around the different membrane groups can be related with the macroscopic response. The dynamics of these structures give place to fluctuations between water populations with different energy and entropy contributions.

## 4. The Membrane as an Open System: Thermodynamic Consequences

The introduction of water as a membrane component changes the definition of a membrane system from a closed to an open one. In consequence, the time-invariant state is not an equilibrium in a closed system in which the maximum entropy or minimum energy is reached when all forces are zero. In contrast, in an open system, the entropy is maintained constant in a time-invariant state (steady-state) in which non null forces are canceled out. In this state, the membrane exchanges water with the adjacent milieu at given surface pressure. An unbalance either of surface pressure or water exchange produces a transient state that the system tries to compensate for.

The approach of Defay-Prigogine in which the interphase is an ionic solution (Figure 1C) is particularly adequate to describe this process and illustrative to extend the role of water in the membrane in accordance with the coupled processes between solute and water named above. According to these authors, when a solute is injected in the subphase of a monolayer, it can diffuse from the bulk aqueous solution to the interphase driven by a concentration gradient. In this process, the water activity at the interphase decreases which produces a coupled flux of water into the interphase. This can explain the increase in surface pressure produced by the insertion of bioeffector in the monolayers at a constant area, and extended to bilayers also when the decrease in membrane density is considered [32,34]. It must be taken into account that this increase in surface pressure is indeed a decrease in the surface tension of the interphase, a point that will be analyzed in detail later.

Consequently, several of the above paradigms concerning permeability and dielectric properties must be revised. Permeability is described by the interrelation of a partition process of the permeant between water and membrane bulk, which is taken as a pure hydrocarbon phase without water, and its diffusion in a homogeneous phase. This is supposed to be valid for single water permeation driven by a permeant solute gradient of concentration across the membrane. A more realistic picture was offered when Thermodynamics of Irreversible Processes (TIP) was applied to explain the permeation of nonelectrolytes. It was concluded that solute diffusion promotes a water flux and that osmosis induces also solute permeation. These conclusions were sustained for the process across the membrane, but no consideration of the changes in water content in the membrane itself was made [35]. This point is discussed in the next session.

## 5. Membrane Hydration and Membrane State

The introduction of the interphase as a binary aqueous solution as part of the membrane structure and the notion of an open system regarding water imposes a different thermodynamic approach to understanding membrane processes.

The thermodynamics of the membrane as an open system is given by the entropy production (dS/dt) that can be written in terms of the sum of forces (X_i_) and fluxes (J_i_) operating on the system defined in two dimensions as in Equation (1):(1)$$\frac{\mathrm{dS}}{\mathrm{dt}}=\sum^{\;}J_{i}X_{i}\;$$

Thus,
(2)$$\frac{\mathrm{dS}}{\mathrm{dt}}=\frac{\Pi \mathrm{dA}}{\mathrm{dt}}+\frac{\Delta {\;\mu \;\mathrm{dn}}_{W}}{\mathrm{dt}}=J_{a}\Pi +J_{w}\Delta \mu =0$$
where Π is suface pressure; $\frac{\mathrm{dA}}{\mathrm{dt}}=J_{a}$ is the change in area (A); Δ μ is the gradient of chemical potential of water and $\frac{{\;\mathrm{dn}}_{W}}{\mathrm{dt}}=J_{w}$ is the water flux.

For the sake of simplicity, the system is taken at constant temperature and in the absence of electric fields and chemical reactions (this can be extended but it is beyond the aim of this work).

When the system is in the steady state, the entropy production is constant and thus dS/dt = 0.

Then,
(3)$$J_{a}\Pi =J_{W}\;\Delta \mu \;$$

As the surface pressure (π) can be expressed by the difference between the surface tension of pure water ($\gamma^{0}$) and surface tension of water with lipids (γ)
(4)$$\pi =\left(\gamma^{0}-\gamma \right)$$
(5)$$\mathrm{Considering}\;\mathrm{that}\;\;\;\;\;\;\Delta \mu \;={\;\Gamma }_{W}{\mathrm{RTlna}}_{W}\;\;\;$$
with $\Gamma_{W}\;$equal to surface water concentration and $a_{W}$ equal to water activity,

It results in
(6)$$\gamma =\gamma^{0}-\Gamma_{W}{\mathrm{RTlna}}_{W}\;\;\;\;\;\;\;\;\;\;$$

In this condition, each value of water activity would correspond to a value of surface tension. In other words, water activity fixes the surface tension, and vice versa, surface pressure (surface tension) would give place to a given water activity.

This analysis denotes the importance of water in the determination of the thermodynamic properties of monolayers and bilayers, mainly its propensity to react in the presence of solute in the bulk phase. According to Equation (6), any process that affects water activity will affect surface tension and hence denoted as a surface pressure increase. Therefore, at a given water activity the membrane surface free energy state is fixed, and thus its propensity to respond to solutes in the bulk adjacencies [22,36].

The point here is to define the properties of water in lipids in terms of its free energy (i.e., surface tension)

## 6. The Surface Domains: Excluded Volume Concept

Cellular membranes are laterally heterogeneous systems that are probably involved in function. The composition of membrane rafts as the archetypical lipid-driven plasma membrane domains has been extensively discussed. The complexity and flexibility of lipid-mediated membrane organization could be functionalized by cells. The different lateral compositional heterogeneity is a ubiquitous feature of cellular membranes on various length scales, conforming molecular assemblies. The nature of the micrometric domains in terms of specific physicochemical properties is a matter of discussion [4,37].

The water content and its organization as H bond networks is concomitant to the composition of lipids and its topology, such as rafts or domains. The mechanical properties to these water regions surrounding the lipid domains acts as a plasticizer along its compressibility modulus [38].

Thus, the friction between lipids and water at this point implies changes in water properties (structure, density, and polarity among others) [33,39] and hence propensity to H bond rearrangements.

These arrangements are affected by the surface pressure (i.e. the area available for the lipids) and therefore its excess free energy of the regions exposed to bulk water changes consequently. Therefore, the propensity of the membrane to “react” to bioeffectors (i.e., biologically relevant compounds in the bulk adjacency) is modulated by water activity and the corresponding surface tension of those regions (Equation (6)).

Different lipid packing states in biomembranes that form coexisting domains (i.e., relatively ordered and disordered domains such as lipids in the liquid condensed or the liquid expanded state) are assumed to have functional characteristics [40]. However, it must be considered that each packing state implies different water levels and arrangements giving place to regions in which water surface tension is different, i.e., the excess of surface free energy. Thus, they act as an energy reservoir that would explain the responsiveness of lipid membrane to bioeffectors present in the adjacent media (aminoacids, peptides, enzymes, etc.). The changes in surface tension, as stated in Equation (6), are modulated by the influence of lipids on water structure and this is manifested in conditions in which lipids form a coherent array (at c.a. 5–10 mN/m) ending at the collapse pressure (c.a. 45 mN/m) depending on the lipid [41]. Thus, a correlation between lipid packing and water activity is inherent to functional activity.

To make explicit the role of surface activity in terms of water activity it is necessary to extend some features of hydration of membranes in analogy to studies made on protein hydration. This is particularly important if the aim is to explain the properties of a membrane in the context of crowded systems [27].

Biological membranes can assume a number of different lipid packing states that may correspond to domains in live cell membranes [42].

Packing is referred to in different structural membrane studies but the thermodynamic properties inherent to it are not rigorously explained. These two aspects are related to surface tension and excluded volume concept as will be explained below.

Excluded volume is a term that represents the unavailable volume of solvent for a solute in a solution. In a simple description, solute dissolves in a salt solution in the water beyond that the hydration shell of the ions. It is also applied to concentrated solutions of macromolecules in which water as a solvent is drastically reduced. This view has been particularly used to refer to the cell as a crowded system [43].

The mutual impenetrability of solute molecules into the hydration shell of the ions and the steric repulsion between ions themselves plays a fundamental role in intermolecular interactions. The activity of a solute depends on the volume that is available for each molecular specie. The steric freedom (entropy) is a function of the size and shape of the solute.

In terms of the bidimensional solution defined as the interphase, the impenetrability in a lipid membrane is given by the polar head group surrounded by the hardcore of hydration molecules (Figure 2) [44].

The hardcore region (blue region) only changes in drastic conditions of dehydration or by the presence of certain compounds that may replace water such as sugars such as trehalose [45,46,47].

Tightly bound water determines the excluded volume of the interphase region contributing to the permeability barrier [48]. In this view, water layers are of a similar magnitude to the hydrocarbon region itself and constitute a repulsion barrier for many permeant solutes and for membrane adhesion [49,50,51,52].

Assuming that the lipid hardcore of hydration is constant, expansion/contraction processes change the water available for solutes in a lipid membrane beyond this hardcore. Thus, the membrane state at a given surface pressure is determined by the relationship of the amount of loosely bound water in relation to the hardcore which is influenced by the presence of the lipids.

In this regard, the thermodynamic response of an interphase (i.e., the change in surface tension) to bioeffectors is related to water organized around the lipids beyond the first hardcore hydration shell. This type of water between lipids beyond the first hydration shell sustains the original model of Defay-Prigogine in which the interphase is a bidimensional solution of hydrated head groups immersed in water [28,32].

## 7. Critical Water Activity and the Cut off Surface Pressure

### 7.1. Critical States: Packing and Cut off Pressure

Recalling that the surface tension of water varies according to the dimension and quality of its environment, it is not difficult to comprehend that the physical chemistry of water confined between lipids (the light blue region in Figure 2) with its hydration hardcore (dark blue) can determine the membrane thermodynamic response.

The typical curves in Figure 3A denote the response of a monolayer at a given initial surface pressure to a bioeffector added to the subphase. The plot of the magnitude of the response (π_f_ − π_ι_) as a function of the initial surface pressure (π_i_) indicates that there is a cut off surface pressure above which no membrane response is observed. This behavior is usually related to the packing of the lipids at the interphase. In a simplistic interpretation, there is no area available to the bioeffector to penetrate the monolayer.

However, the insertion process is not driven by the area but by free energy changes. Therefore, other properties concerning the interphase must be relevant.

Packing is frequently taken as the equivalent of the ordered state. However, packing is referred to lipid molecules i.e., the special arrangement in terms of distance. Such ordering is ascribed to lipids immersed in water since properties are assumed to be measured in fully hydrated membranes. The packing increased by compression enhances the hydrophobic interactions of the monolayer or bilayer promoting a decrease in water order. Therefore, the two concepts must be compatibilized with each other.

Packing is a geometric criterium and order-disorder is a thermodynamic one. However, the geometrical criterium to some extent can be translated into thermodynamic terms if the energy contribution is considered in terms of the excluded volume as described above.

### 7.2. Cut off and Critical Water Activity

The surface pressure can be expressed in terms of water activity (a^i^_w_ according to equation π = −C Γ_w_ RT ln (a^i^_w_), where Γ_w_ is the water concentration at the interface and C is function of the frictional coefficient lipid-lipid, lipid water and water lipid [33]. Thus, the cut-off pressure reflects a critical water activity.

At the cut-off pressure, water activity beyond the hardcore is zero and no water domains are formed. Thus, no effect of bioeffectors is found. This is produced at the critical water activity (Figure 3D) above which water domains are formed (Figure 2B) and the excess free energy triggers the bioeffector perturbation [32]. The water domains are characterized by a surface tension fixed by the water activity according to Equation (6).

For a given water activity at the membrane, the perturbation (ΔΠ) depends on the type of bioeffector, and of lipids and it may have different consequences. Among them, drastic changes in the dielectric permittivity of the membrane maintaining the bilayer conformation allow permeation of polar and ionic solutes, a process that cannot be explained by the classical non-polar slab of the bilayer in classical models [28,53]. Another consequence is the induction of non bilayered structure abandoning the bilayer conformation [54].

**Figure 3 molecules-27-04994-f003:**
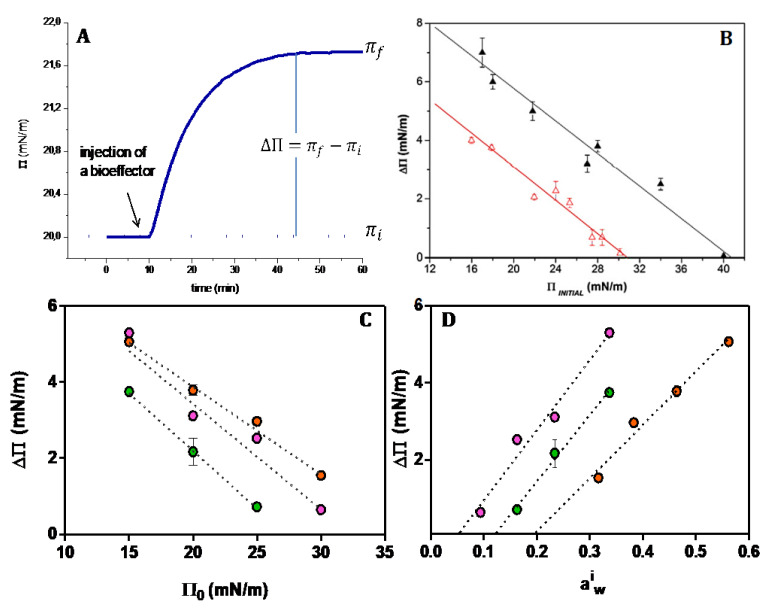
Perturbation of a responsive membrane to the different bioffectors. (**A**) Typical changes of the monolayer surface pressure upon the addition of a bioeffector. (**B**) Difference of the final and initial surface pressure for different initial surface pressures induced by a soluble protease as a bioffector on DMPC (∆) and DMPE (∆) monolayers (taken from Ref. [55]). (**C**) Difference of the final and initial surface pressure for different initial surface pressures induced by chlorogenic acid on DPPC (●), Di 16:0 ether PC (●), DMPC (●) (taken from Ref. [56]). (**D**) Difference of the final and initial surface pressure as a function of the initial water activity in the monolayer calculated according to π = −C Γ_w_ RT ln (a^i^_w_) for DPPC (●), Di 16:0 ether PC (●), DMPC (●) (See text).

Compression produces the collapse of the lipid monolayer at high pressures. In this stage, lipids are packed to a limit in which the hard core first hydration layers are in contact (Figure 2A). This is the minimum excluded volume (blue spheres) which implies a volume non available for solutes. In a ΔΠ/Π curve as in Figure 3B,C, this value is reflected by the critical cut-off, i.e., the pressure at which no response is observed. The response is observed when the initial surface pressure is below the cut off, which is somewhat below the collapse pressure. In consequence, the water activity at the cut-off is higher than that at the collapse (Table 1).

This means that the excluded volume sensitive for membrane response is larger than the hardcore water inferred from the collapse pressure. This implies that the geometrical space created is much lower than the molecule volume of the bioeffector. Therefore, the insertion of the biocompounds can be more suitably explained in terms of energetic considerations.

## 8. Hydration and Phase Transition

The increase in surface pressure produced by a bioeffector is a measure of the decrease in surface tension. This means that the initial surface tension, which reflects a particular free energy state of the membrane, is reduced and this can only be produced if initial water arrangements have certain level of potential energy. These arrangements should be reflected in the thermodynamic properties of the membrane.

Being the membrane composed of two slabs of a strongly different physical chemical nature, the main phase transition should be an emergent of the system.

Heat causes a global thermal transition due to the interplay of molecular interactions between the nonpolar and polar regions with water playing a fundamental role.

Considering the presence of water, the total enthalpic change in the transition can be written as
(7)$$\Delta H=\Delta H_{f}+\Delta H_{l}\;$$
where ΔH_f_ is the enthalpy of fusion of the hydrocarbon chains and ΔH_l_ is the energy involved in the ionic network in the aqueous excluded region. In turn,
(8)$$\Delta H_{l}=E_{r}+\Delta H_{h}\;\;\;\;\;\;\;\;\;\;$$
where E_r_ energy of fusion of the ionic (polar groups) lattice and ΔH_h_ the enthalpy of hydration of the head groups.

ΔH_f_ and E_r_ are endothermic processes and ΔH_h_ is an exothermic one. Thus, how endothermic ΔH is, would depend, to a great extent, on the hydration of the polar groups [57].

Thus, the lipid propensity for membrane responsiveness involves water rearrangements reflected in the ΔH values.

## 9. The Limits of the Membrane in Crowded Systems and Signal Propagation

The limits confining the interphase are not sharp. Water may penetrate the first 4C atoms, and the presence of the polar groups may affect layers of water in the bulk solution. In terms of Molecular Dynamics (MD) this may have an extension of around 12 Å [58]. This implies water facing different kinds of molecular residues and thus with different free energy content with different capabilities to exchange.

There is a great content of ambiguity in the definition of the cell limits and cell membranes: boundaries become in barriers and these in membranes, which finally are resumed in bilayers giving to them the same hierarchy.

This confusing nomenclature is due, in our view, to macroscopic observations disregarding the definition of interphases as a region separating two well-characterized (continuous) regions.

A great deal of care has been taken to establish the limit between the hydrocarbon core and the aqueous phase since Luzzati et al. [59] interpreted the X-ray diagram of lipid suspensions, considering the former proposal of the lipoidal nature of the cellular boundary by Overton [60].

Over the years, many assumptions have been made in order to calculate area per lipid from measurements of interbilayer distance and membrane thickness [23,49,61,62]. The membrane thickness also includes the polar head groups. Needless to say, in these calculations a simplification is dragged, e.g., the density of water (molar volume) is constant and similar to the bulk [63].

Further, with the aid of molecular dynamics and new scattering methodologies, the limit between water and the hydrocarbon region has become diffuse. Water can extend into the first 4–5 C atoms below the carbonyl groups and from the plane of the phosphates and the different polar residues attached to it, toward the external water. Consequently, the neat differentiation of water near the groups and the bulk is gradually diluted [49,58,64]. In spite of these uncertainties, the phase properties of the membrane have been mostly interpreted in terms of the properties of the non-polar region. However, the only region that can be considered properly as a hydrocarbon phase is that at the terminal -CH_3_ residue in the center of the bilayer. That is why most of the interpretations lay on the idea that the phase transition is due to the fusion of the hydrocarbon chains that are affected by water, but no consideration of the thermal properties of the water region itself is made [65]. However, in the phase transition water structure also changes and this change is implicit in dynamic process [57].

The behavior of water beyond the hardcore (blue light region in Figure 2B) at the phase transition is shown in Figure 4.

The data in Figure 4 clearly indicates that the thermodynamic properties of water between lipids are different in comparison to bulk water, since a clear break is observed in the presence of lipids that is absent in pure water. The main observation is that the molar fraction of water monomers increases (part A) parallel to the water tetramer decrease (part C), indicating an order-disorder transition. In both cases, a break is observed at the phase transition followed by the phospholipid groups (red dots) that is not present in pure water (black dots). Thus, the transition in water associated with the membrane reflects the membrane state.

If a living cell is represented by a water solution of proteins and ions enclosed by the membrane as a bilayer, the general criterium is to take water inside having no major differences from normal liquid water bathing the cells. In contrast, from the colloidal point of view, cytoplasmic water is stabilized, essentially continuously, by interactions with proteins, forming an extensive, ordered lattice in which the water molecules have reduced translational and orientational freedom and ordered microscopic arrangements compared with free water [64]. However, the so-called “bound” water invoked by the Ling’s AIH can not only be ascribed to the presence of macromolecules in the microscopic heterogeneity of the cytoplasm (due to the influence of cell organelles) but also to the presence of lipid aggregates such as a membrane [65,66,67]. These concepts may be incorporated into membrane theories without entering into conflict with Ling’s theory.

It is precisely the dynamics of lipid-water and water-lipid association that make it possible to think in a dynamic, coherent coacervate responding to and driven by cell metabolism. Moreover, the propensity of lipids to stabilize in different topological and conformational arrays beside lamellas seems to be driven by lyotropic transitions, i.e., water activity variations at a constant temperature [68,69].

The view of including water as a component of membrane structure inquires the concept of bilayer as a permeability barrier. In addition, it introduces water as a participant in the lipid conformational changes.

Despite this, it is quite reasonable that lipids in that context can suffer lyotropic phase transitions maintaining the bilayer conformation and/or polymorphic changes abandoning the lamellar conformation, in such a way that the bilayer, as usually pictured, is not a continuous barrier [13,70].

Lipids interact between them by short-range (weak) forces that make them considered a soft material. However, lipids have specific sites of hydration (phosphate-PO_2_^−^ and carbonyl-CO groups) and acyl chains that impose a particular ordering of water molecules. Moreover, the correlation of polar-polar and chain-chain interactions are modulated by water content giving place to cooperative phenomena as described elsewhere [57,71,72,73].

In this regard, following the TIP approach [28,33] it is important to point out that water content may be modulated by osmosis, and this should not be considered as a process to produce a flux across the membrane (such as in the classical membrane theory) but as one in which water is extruded from the membrane itself. That is to say, it may cause lyotropic transitions. Even in the case that the bilayer is preserved, permeability is a much more complex process than that described by partition and diffusion in the classical membrane theory. This is so because the partition is not uniform along the membrane thickness, and diffusion is not given by a constant coefficient as described by Fick’s law [32]. In contrast, it varies along the process due to the relaxation of the water structure. Different water activities can give place to changes in the lipid conformation different from bilayers in extreme cases of dehydration [28].

## 10. Unifying Membrane Approach with Water Colloid Systems

If the membrane is taken, thermodynamically speaking, as an open system, it is thought that exchanges of matter with the surroundings are allowed. The classical idea of close systems is that membrane material is constant. The exchange of water through the membrane barrier (transient or non-stationary water) is explained without altering its composition and even maintaining constant its density (packing) along the process. However, density can change maintaining the lipid constant, at surface pressures well below the collapse, because the water content is modified. This is more noticeable when water is dragged by a permeant. Water dissolves in kinks affecting the density of the lipid matrix [41,57,72] and its dielectric properties due to solute penetration [53,74]. In this sense, both, open and non-autonomous systems are interrelated concepts.

In a non-autonomous system, the limit of the aqueous layer is not determined by the membrane itself but by the media which is in contact with. Thus, if the membrane faces a colloid aqueous one, such as cytoplasm, it will have different states than if the media is pure bulk water.

In terms of compartmentalization, continuous semipermeable membranes are not required as the FMM model does. Compartmentalization can be taken as a kinetic process and not necessarily as a physical barrier. Different kinetics of solute penetration were found at different cholesterol ratios in lipid monolayers and constant areas and at different surface pressures, all of them related to changes in water activity in the interphase [22,41,75].

It is possible that the presence of lipids with the propensity to stabilize in non-bilayer ensembles strongly affects the kinetics of permeation. The proposal that the dependence of the conformation of the lipids with the state of the internal cytosol is an extension of the concept of an open (non-autonomous) system, since it considers a structural change in the lipid arrangement much more drastic than the lipid density changes in bilayers. This would deserve further analysis within the frame of Thermodynamics of Irreversible Processes (TIP) [33].

## 11. The Thermodynamic Response: Membrane State/Hydration State/State-Function Relationship

As said, the concept of the open/non autonomous phase can be framed in terms of Gibbs free energy. J.W. Gibbs was a visionary. It is not known if he was conscious of that, but the extension of the physical thermodynamics to chemical systems is one of the most relevant theoretically based experimental formalisms on which science is based today [76,77].

The free energy decrease can be enthalpically or entropically driven. If water is the messenger in order to make coherent the biochemical processes in cells including lipids, the two aspects are fundamental. First, the energy exchange is driven by the energy of hydrogen bond networks (water-water, water-lipid interactions) and by the entropy change due to the unique properties of the structure of water itself. An example of this is the water exchange between monomers and tetramers shown in Figure 4.

The important observation that the periodicity of glycolytic oscillations and that the attendant coupled oscillations in water relaxation are slowed down by deuterium oxide (heavy water) makes it reasonable that this is a consequence of the stronger energy of the D-bonds in comparison to the H-bonds [54]. Thus, this increase in stability enhances hydrophobic interactions and dampens oscillations.

Therefore, unstable or metastable systems have the root in the H bond strength of water with itself and with the walls in a restricted environment [64,77,78,79]. Therefore, more details in relation to hydrogen bonding energy in different spatial configurations are needed.

### Responsive Structures and H Bond Networks

The presence of a cellular component governs the emergent properties that would affect the cellular interior dynamically. This implies that water organization mainly by hydrogen bonding should be higher, in terms of extension and stability, than bulk water. If this is so it should be reflected in dielectric relaxation and NMR results [53,78]. However, no more than 10% of cell water is supposed to be highly structured. The possibility of highly organized water was thought to be confirmed when the polywater was apparently discovered but was promptly demonstrated as a failure [80,81]. In addition, if the highly structured water exists the strength of H bonds would damp oscillations as occurs in D_2_O as denoted above [54].

However, the new perspective of introducing water as a structural/functional component in cytoplasm and membranes can be reasonably accepted if interphase phenomena are considered to play a key role [82]. This is more logical when the cell interior is thought of as a crowded system, in comparison to a broth of protein ionic solution. In this regard, volume chemistry should be replaced by surface chemistry [27,83,84].

Lipids are part of the complex and crowded system and may determine kinetic and relaxation phenomena that are not restricted to permeation across the membrane. The relaxation implies the reorganization of water arrangements and therefore, changes in polarization, density, and compressibility. These are noticeable properties of water derived from hydrogen bonding network with noticeable plasticity [38].

The relaxation of water in the vicinity of lipid head polar groups measured with fluorescence probes in cellular aggregates of lipids is a strong indication that responsive behavior is due to water at the *interphase* which may be coupled to the cytosol [85,86,87]. The physical coupling can be a combination of properties emerging from water arrangements by H-bonds. As observed by FTIR, water bands are modified by the presence of lipids in different states (gel or liquid crystalline). These bands change at the phase transition mirroring the phase transition in the lipid phase [57]. The changes in bands are a consequence of the evolution of water populations from tetrahedral array (4H bonds) along 3, 2, 1, and 0H -bonded species [64]. The transition in the water populations “resonates” with the lipid state, and probably with the interior cell structure and central metabolism as well [88,89].

The strong cohesion in the cytoplasm can be extended not only between proteins and water but also between lipids and water. This statement fits with the association claimed in the association-induction hypothesis.

The living state is defined as a *cooperative* state indicating that each state is well defined and discrete and that there are neighbor-to-neighbor interactions among the individual elements. These features have been found in lipid molecules in the processes of adsorption of aminoacids and proteins to lipid membranes [90].

The cooperativity gives place to propagation. Polarized multilayers of water are not only in proteins but also in lipids and its propagation from and to the interphase is a property of the H bonds. According to AIH theory, fixed charges and associated counter-ions are separated by up to three dielectrically saturated water layers. This is comparable to the thickness of water layers in the membrane interphase polarized by charges such as phosphate and carbonyl groups, constituting the excluded volume which contributes to the repulsion forces and the responsiveness of the membrane (see Figure 2 and Figure 3) [28,32,50,58]. The complete restriction of reorientation applies to the first hydration layer of these groups. However, the shells beyond it and in between the acyl chains can be modulated by surface pressure, mechanical constriction imposed by the water activity [65,91]. Membrane expansion-contraction may be induced by intrinsic processes of the cell driven by metabolic activity or by mechanical stress imposed by the external environmental conditions (osmotic swelling and shrinkage) [92].

The dynamic behavior of water at a constant temperature can be of electrical nature, more precisely derived from the dielectric properties implied in the polarization measured with fluorescent probes.

The dielectric permittivity is, in turn, much higher in ice than in liquid water explained by the higher degree of association of water molecules in a periodic array of tetrahedral coordination (4H-bond population predominates).

The dielectric permittivity is due to the orientation in an electric field of the dipoles. The common picture is that the reorientation implies the breaking of hydrogen bonds. However, a complete rotation of the dipole can be achieved by displacement of the H along with the H bond that remains unbroken. The water dipolar relaxation (rotational dynamics) is represented schematically below (Figure 5).

The inversion in the direction of the water molecular dipole (arrows) can be conducted without rotation of the molecule (i.e., without breaking H bonds), but with a displacement of the protons along with the linear H bond. The extent of the propagation will be limited by the density of states of 4, 3, 2, and 1-coordinated molecules by H-bonds giving a multiplicity of polarization states. Thus, polarization at the interphase and the propagation to the cell interior can be due to H displacement as in the Grothuss mechanism.

Moreover, if water populations change as a consequence of changes in lipid phase states or internal cell processes, a complex (and fast) response of polarization can result. The lack of water would alter this propagation essential for the living state [88,89].

Thus, “dynamic interphase hydration” can be controlling the probability of lyotropic mesomorphic transitions, allowing lipid self-assemblies and metabolically reactive structures in the cell to be coupled.

## 12. Concluding Remarks: Membrane Hydration and Membrane State. An Alternative Way to Membrane Response

The consideration of the interphase as a bidimensional hydrated polar head group solution coupled to the hydrocarbon region whose polarity varies according to its water content deserves a new thermodynamic interpretation.

In this regard, the consideration of the excluded volume in the free energy of the lipid interphase is useful to have a picture of the membrane in crowded and confined systems.

Thus, the description and characterization of crowding- and confinement-induced effects in living organisms, which have been focused on considering proteins in the cell cytoplasm, can be extended by the inclusion of the membrane itself as a crowded system and in connection with the cell interior.

The presence of water in the first 4C atoms and around the polar groups affects the water properties and implies water with different free energy content in comparison to pure bulk water, and hence with different capabilities to react.

Excluded volume can be modified by different chemical and physical conditions, such as surface pressure, osmosis, mechanical forces, and membrane components (e.g., unsaturated lipids and cholesterol) that may alter the lipid water ratio. This is the basic criterium to describe the membrane state as an open, non-autonomous and responsive system as it has been discussed [28,33].

The concept of an open/non-autonomous membrane system is visualized in the interrelationship between metabolic events and membrane polymorphic changes.

Therefore, the AIH and lipid properties interplay should consider hydration in terms of free energy modulated by water activity and surface (lateral) pressure.

This review is an attempt to incorporate the membrane as a structural component following the AIH and coacervate proposal.

In a recent review, the breakdown of current paradigms in the bilayer knowledge has been analyzed considering water as a fundamental structural and thermodynamic component of membrane systems [28,33]. This proposal can be extended to lipids organized in non-bilayer structures.

This criterium may also contribute to linking the role of lipids with Ling’s cell theory, giving the water a central role from the thermodynamic viewpoint.

Thus, the challenging step forward is to take water organization at the limits of the membrane as a structural, dynamic, and functional element in living cells that may penetrate lipid membranes [32,90]. In this view, the new physical-chemical tool includes Thermodynamic of Irreversible Process formalisms and their dissipative and non-dissipative components to comprehend complex systems, such as cells and cell structures [28,87,92].

## Figures and Tables

**Figure 1 molecules-27-04994-f001:**
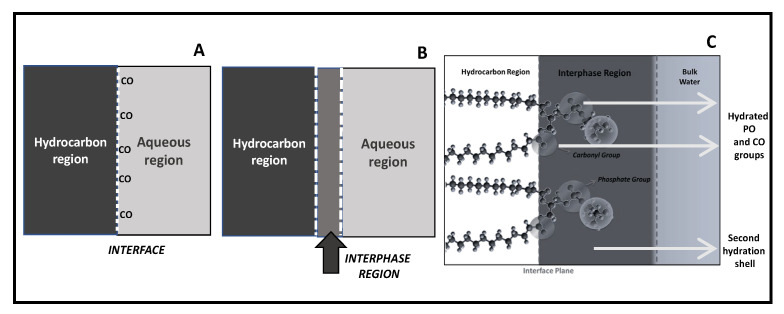
Different schematical descriptions of the interface: (**A**) The Gibbs model, (**B**) The Guggenheim model and (**C**) The membrane interphase model.

**Figure 2 molecules-27-04994-f002:**
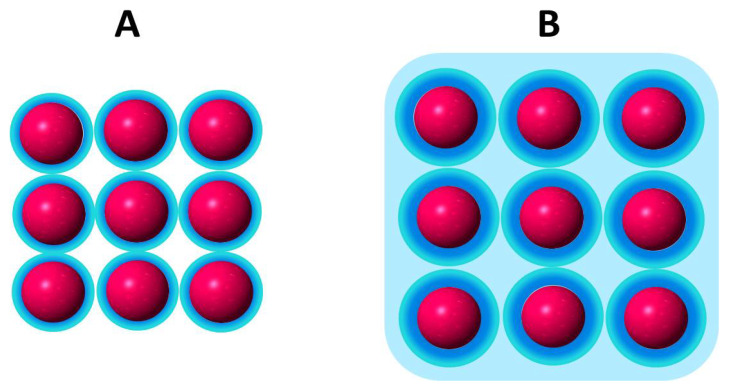
Schematic representation of the excluded volume in the head group region (red spheres) of a lipid membrane. (**A**). Blue region: hardcore first hydration shell. (**B**). Light blue region: water available as solvent (confined water) although of different properties than bulk water.

**Figure 4 molecules-27-04994-f004:**
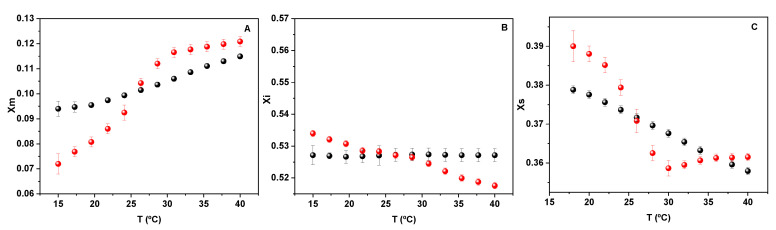
Molar fraction of water monomer (**A**) water bound with less than four H bonds (**B**) and tetracoordinated water molecules (**C**) as a function of temperature. Black symbols correspond to pure water, red symbols correspond to water in the presence of lipids.

**Figure 5 molecules-27-04994-f005:**
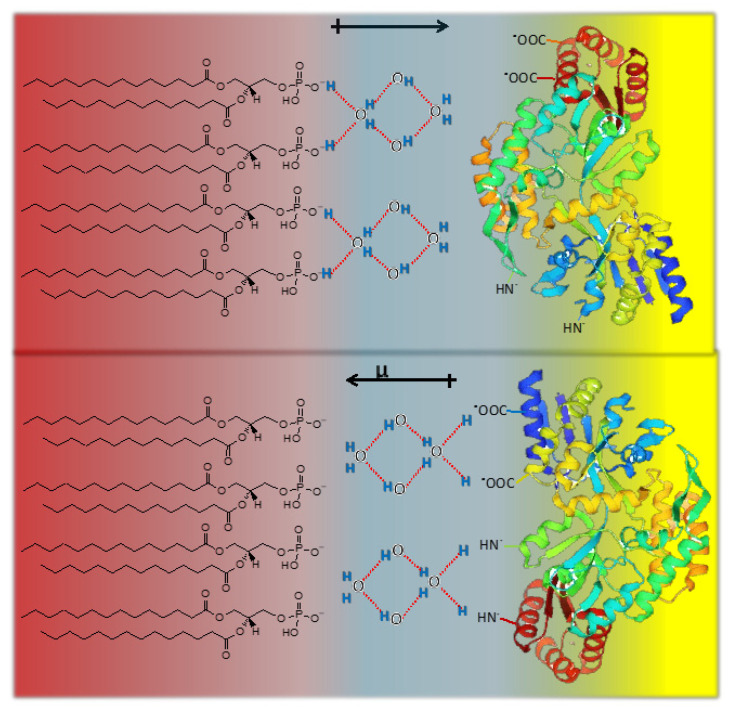
Inversion of dipole direction by displacement of protons without molecular rotation. Protein conformation may change inducing a water polarization by proton displacement that propagates to the membrane interphase and vice versa.

**Table 1 molecules-27-04994-t001:** Comparison of surface pressure and the corresponding water activities at the collapse and cut off pressures [55,56].

Type of Lipid	Bioeffector	π_saturation_ ± sd (mN/m)	π_cut off_ ± sd (mN/m)	a_w_ (Saturation)	a_w_ (Cut Off)
**DPhPC**	Aqueous protease	48.0 ± 0.7	39.6 ± 0.4	0.308	0.378
**DOPC**	Aqueous protease	47.2 ± 0.9	41.4 ± 0.3	0.314	0.362
**DMPC**	Aqueous protease	47.8 ± 0.8	41.5 ± 0.5	0.309	0.361
**DPPC**	Aqueous protease	46.6 ± 0.6	39.5 ± 0.9	0.034	0.057
**DMPE**	Aqueous protease	45.0 ± 0.5	30.6 ± 0.1	0.038	0.108
**Di(ether) PC**	Aqueous protease	48.0 ± 0.7	31.9 ± 0.3	0.307	0.457
**Di(ether) PE**	Aqueous protease	44.5 ± 0.5	29.4 ± 0.6	0.040	0.118
**DMPC**	CGA	47.8 ±0.8	36.9 ± 0.3	0.160	0.243
**DPPC**	CGA	61.4 ± 1.3	27.2 ± 0.3	0.011	0.134
**16:0 dieter PC**	CGA	58.5 ± 1.1	32.4 ± 0.1	0.013	0.092

## Data Availability

Not applicable.

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
