# Peer review of "Water as a Link between Membrane and Colloidal Theories for Cells"

_molecules, 2022, doi:10.3390/molecules27154994_

Round 1

Reviewer 1 Report

The presented manuscript concerns the role of water in biomembranes.  The importance of water molecules at the interface between membrane and bulk water is discussed in detail. The Authors propose that the role of interface between the membrane and bulk water could form a link between classical membrane cell models and colloidal ones. The detailed discussion of interfacial water, especially in terms of excluded volume, on membrane permeability is very interesting. The treatment of phospholipid membrane as an open system that is able to exchange water with its surroundings, and interfacial water as essential component of biomembrane is able to link the membrane and coacervate theory.  This approach is very interesting and could lead to further understanding of living cell as a confined system in which confinement induced effects play crucial role. Overall I recommend the publication of the manuscript after some minor revision. 

Following issues should be addressed before publication:

1. Uniform subsection numbering should be introduced in the manuscript, as for now different symbols are used: 1), 2.-

2. I would suggest numbering all equations in the manuscript. All symbols used in the equations should be explained at the first appearance in equation. Symbols that appear in section 4 are introduced in section 6, this should be amended.

3. Color of symbols presented in Fig. 4 should be explained also in figure caption. 

4. TIP acronym introduced without further development in section 3. Full name is provided in section 10.

Author Response

Numbering of subsection has been corrected.

equation were numbered sequently and the meaning of the term explained appropriately.

symbols in figures were added in the figure caption.

Acronisms were clarified at the beginning of the text.

Reviewer 2 Report

This is an interesting paper postulating a water role in the potential confluence of two theories regarding the concept of the cell as a physical system. The manuscript is well written and of value.

Addressing some issues would improve the manuscript quality:

1) Please extend a bit the abstract to include the most relevant two or three ideas included in the "concluding remarks".

2) Section 5: the role played by water in membrane functionality in regard to lipid phase/state as well as its thermodynamic contribution (concluding remarks) are not clear. Explain better these concepts.

3) Table I: use always the same amount of decimal digits. Change point instead of comma decimals.

4) Some typos and grammar mistakes should be amended: "resumed", "being this of similar properties than bulk water",  ...

Author Response

The abstract was modified considering the y conclusions reached in the manuscript.

A new paragraph was added in section 5 ( page 11). 

Table was redone.

English typos were amended.